# Microplastic Distribution Characteristics and Sources on Beaches That Serve as the Largest Nesting Ground for Green Turtles in China

**DOI:** 10.3390/toxics12020109

**Published:** 2024-01-28

**Authors:** Ting Zhang, Deqin Li, Yunteng Liu, Yupei Li, Yangfei Yu, Xiaoyu An, Yongkang Jiang, Jichao Wang, Haitao Shi, Liu Lin

**Affiliations:** 1Ministry of Education Key Laboratory for Ecology of Tropical Islands, Key Laboratory of Tropical Animal and Plant Ecology of Hainan Province, College of Life Sciences, Hainan Normal University, Haikou 571158, China; tingzhang1994@163.com (T.Z.); 18575557896@163.com (D.L.); 18722945204@139.com (X.A.); 15120643775@163.com (Y.J.); wjc@hainnu.edu.cn (J.W.); haitao-shi@263.net (H.S.); 2Hainan Sansha Provincial Observation and Research Station of Sea Turtle Ecology, Sansha 573100, China; 13707506382@163.com (Y.L.); 18789047321@163.com (Y.L.); zeek1115@aliyun.com (Y.Y.); 3Marine Protected Area Administration of Sansha City, Sansha 573100, China

**Keywords:** green turtles (*Chelonia mydas*), North Island, microplastics, plastic fragmentation

## Abstract

The threat of microplastics to marine animals and habitats is increasing, which may affect sea turtle nesting grounds. The Qilianyu Islands are the largest remaining green turtle (*Chelonia mydas*) nesting grounds in China. Despite being far from the mainland, microplastic pollution cannot be ignored. In this study, the level of microplastic pollution in surface sediments from three different zones, namely, the bottom, intertidal, and supratidal zone, was investigated on North Island, Qilianyu Islands. The results showed that the abundance of microplastics in the supratidal zone was significantly higher than that in the bottom zone and intertidal zone (r = 3.65, *p* = 0.011), with the highest average abundance of microplastics located on the southwest coast of North Island. In the bottom zone, only plastic blocks (88%) and fibers (12%) were found. The main types of microplastics in the intertidal and supratidal zones were plastic blocks (48%) and foam (42%), with polyethylene (PE) (40%) and polystyrene (PS) (34%) being the predominant components. These types and components of microplastics differed from those in the surrounding seawater, but corresponding types and components were found in the plastic debris on the beach. Meanwhile, it was also observed that there were multiple instances of fragmented plastic on the beach. Thus, we suggest that the microplastics on the beach in North Island were mainly derived from the fragmentation of microplastic debris, indicating secondary microplastics. It is recommended to further strengthen the regular cleaning of plastic debris on the beach, especially the removal of small plastic debris, in order to reduce the pollution from secondary microplastics generated by the fragmentation of beach plastic debris and to better protect China’s most important sea turtle nesting site in the South China Sea.

## 1. Introduction

At present, the use of plastic products has become essential for human life [1]. Plastic’s great variety of uses has caused a continual increase in its consumption; whereby, approximately 300 million tons of plastic are produced each year, of which 13 million end up in rivers and oceans [2]. Microplastics are plastic particles with a diameter smaller than 5 mm. Due to this small size, they are difficult to clean up in the environment and have a wider distribution. Research has shown that plastic pollution is ubiquitous, from remote corners of the Earth to high mountain lakes and deep-sea sediments [3,4,5]. The physical and chemical properties of microplastics are relatively stable, allowing them to persist in the environment for hundreds or even thousands of years. Additionally, organic pollutants and inorganic pollutants, i.e., trace metals, can also be found adsorbed onto the surface of microplastics, posing persistent threats to ecosystems [6,7].

Besides some natural sources, microplastic particles are generally formed upon the breakdown of plastic debris through processes such as weathering, oxidation, ultraviolet radiation, and biodegradation [8]. The plastic debris present on beaches can further break down into microplastics due to long-term exposure to ultraviolet radiation and abrasion with sand particles. With the accumulation of time, the size of microplastics becomes smaller and their abundance increases [9,10]. Studies have shown that the presence of microplastics on nesting beaches has negative impacts on turtle hatchlings [11,12]. The determination of turtle gender depends on incubation temperature [13], and microplastics on beaches can cause an overall increase in beach temperature, affecting turtle incubation temperature and leading to gender imbalances [14,15]. Furthermore, microplastics often contain harmful chemical pollutants such as heavy metals and organic compounds. These pollutants can affect embryo development through permeation, reduce hatching success rate, and ultimately threaten the sustainability of turtle populations [16,17].

Sea turtles are umbrella species in marine ecosystems, and their conservation is of great significance for maintaining marine biodiversity, ecological balance, and promoting harmonious development between humans and nature [18,19]. The research and conservation of sea turtles have received much attention from in China, and currently, all sea turtle species have been listed as level-I protection on the “List of Wildlife under Special State Protection” [20]. The green sea turtle is the only sea turtle species that nests in Chinese waters, and its population accounts for 87% of the five sea turtle species in the South China Sea [21]. Furthermore, green sea turtles breeding in Xisha Islands are a new geographic population with important research and conservation value [22,23]. The Qilianyu Islands (northeastern Xisha Island) are currently the largest nesting site for green sea turtles in China. Therefore, it is crucial to monitor the pollutants in the nesting grounds of sea turtles in this area and conduct source tracing analyses for timely restoration and improvement.

Our previous studies showed the presence of plastic debris and microplastic pollution in the nesting grounds of green sea turtles in the Qilianyu Islands of the Xisha Islands [24,25]. However, the spatial distribution characteristics of microplastic pollution and its sources in this area are still unclear. Therefore, this study systematically evaluated the current status and causes of microplastic pollution in the sea turtle nesting grounds by comparing the different abundances of microplastics in the bottom zone, intertidal zone, and supratidal zone, as well as the different abundances in the north coast, southwest coast, and south coast of North Island, Qilianyu Islands. Additionally, we compared the different compositions among microplastics on the bottom, intertidal, and supratidal sediment, and microplastics in seawater, in order to determine the main pathways for microplastic pollution in the nesting grounds of North Island.

## 2. Materials and Methods

### 2.1. Sample Collection

In the green turtle nesting grounds of North Island in the Qilian Islands, three sampling transects were established on the south coast, southwest coast, and north coast of North Island. The geographic coordinates of the sampling points were recorded using the global positioning system (GPS). During the southwest monsoon period in May 2021, surface sediment samples were collected from three zones, including bottom zone, intertidal zone, and supratidal zone, at each sampling point. Due to the large waves during the northeast monsoon period in October 2021, only surface sediment samples from the intertidal zone and supratidal zone were collected at each sampling point. The sampling area for each point, excluding the seabed mud, was 25 × 25 cm, with a collection depth of 0–2 cm [11].

### 2.2. Sample Separation

The continuous flow flotation device established by [26] was used for microplastic separation. The collected samples were divided into 5 portions, one portion was used for microplastic extraction, and another portion was placed in an electric constant temperature drying oven for drying, and the sediment moisture content was calculated after drying. For each sample, 250 cm^3^ of dry sand was placed in a beaker. The preliminary separated substances were obtained through flotation with saturated sodium chloride solution (density 1.2 g/cm^3^), and then the supernatant was collected through secondary flotation with sodium iodide solution (density 1.4 g/cm^3^), to ensure the recovery of microplastics [25,26]. The collected supernatant was mixed with 30% hydrogen peroxide solution (the mixing ration of supernatant and hydrogen peroxide was 1:3) for 24 h and then filtered using a vacuum filtration device through a 0.45 μm glass fiber membrane (GF/F, 47 mm Ø, Whatman, Shanghai, China). The retained filter membrane was placed in a clean glass petri dish for further analysis [27].

### 2.3. Characterization of Microplastics

The microplastics on the filter membrane were identified, photographed, and counted using a binocular stereomicroscope (SZX-10, METTLER TOLEDO, Shanghai, China). When inspecting white or transparent particles, extra caution was taken, and tweezers or the addition of HCL were used for further confirmation. Based on their morphological characteristics, the microplastics were classified into five categories: foam, plastic blocks, fibers, microbeads, and films. According to the color differences of the microplastics, they were further categorized into black, white, yellow, green, gray, blue, and other colors. Nano Measure 1.2 software was used for counting, and the particle size was measured based on the length of the longest side of the microplastics [28]. Additionally, the particle sizes were divided into five categories: 0.05–1 mm, <1–2 mm, <2–3 mm, <3–4 mm, and <4–5 mm [27].

We selected a representative subset of microplastics from each group, and their surface structure was tested for polymer type using a Fourier transform infrared spectrometer (IRTracer-100, SHIMADZU, Kyoto, Japan). Microplastic particles were placed on the sample area, and during data acquisition the ATR imaging attachment was in direct contact with microplastics on the filter membrane. The detector spectral range was 600–4000 cm^−1^, co-adding 16 scans at a resolution of 8 cm^−1^. The spectra were processed using Lab Solutions IR software and compared with the IR polymer spectra library. When interpreting the Fourier transform infrared spectroscopy (FTIR) output, only the readings with confidence levels of 70% or higher were considered reliable and accepted (after visual inspection). All confirmed polymer types were included in our results.

### 2.4. Experiment Quality Control

All containers were rinsed at least three times with Milli-Q water and then dried before the start of the experiments. All plastic equipment was replaced with non-plastic types if possible. If this was not possible, they were rinsed three times with Milli-Q water and then inspected to ensure that no plastic blocks were generated during sample processing. In addition, all containers were always covered with aluminum foil to avoid contamination. Nitrile gloves and cotton lab coats were worn throughout the experiment, with lab windows remaining closed [25]. Three procedural blanks were set to minimize contamination from the environment throughout all sample pretreatment and identification steps, and the results showed that no microplastic particles were detected.

### 2.5. Data Processing

Statistical analysis was performed using SPSS 19.0 software and the ‘stats’ and ‘rstatix’ packages in the statistical software R v. 4.2.2. The relevant data in the study were expressed as mean ± standard deviation (Mean ± SD). *p* < 0.05 was considered statistically significant, and *p* < 0.01 was considered highly statistically significant (two-tailed test). Following the microplastic survey method for sea turtle nesting beaches by [11], the average microplastic abundance in the intertidal zone and the supratidal zone at the same location were considered as the microplastic abundance for that specific location.

## 3. Results

### 3.1. Abundance of Microplastics in Beach Sediments on North Island, Qilianyu

In the intertidal zone and supratidal zone of North Island, a total of 4333 microplastics were collected during two monsoon seasons. The overall abundance of microplastics on North Island was 1513 ± 170 pieces·m^−2^ and 286 ± 43 pieces·kg^−1^. The comparative results showed that the abundance of microplastics on North Island was much lower than that in coastal beaches in Guangdong and Hong Kong, and was similar to that of some islands in the Xisha Yongle Islands (Table 1).

### 3.2. Spatial Distribution of Microplastics in Beach Sediments on North Island, Qilianyu

The average abundance of microplastics in the bottom and intertidal sediment of the Northern Island was 277 ± 102 pieces·m^−2^ and 687 ± 169 pieces·m^−2^. The abundance of microplastics in the supratidal zone was 2339 ± 385 pieces·m^−2^, which was significantly higher than that in the bottom and intertidal zone (r = 3.65, *p* = 0.011) (Figure 1).

The abundance of microplastics on the southern coast of North Island was 1428 ± 100 pieces·m^−2^. The abundance of microplastics in the southwest coast was 1777 ± 138 pieces·m^−2^. The abundance of microplastics on the northern coast was 1334 ± 141 pieces·m^−2^. The abundance of microplastics in the southwest coast was higher than that on the southern and northern coasts (Figure 2).

### 3.3. Morphological Characteristics of Microplastics in Beach Sediments of North Island

The separated microplastics exhibited diverse morphological characteristics (Appendix A). Among them, only plastic blocks and fibers were found in the bottom sediment, accounting for 88% and 12%, respectively. The main types of microplastics in the intertidal zone and supratidal zone were foam and plastic blocks, but their proportions were different. Foam accounted for a higher proportion of microplastics in the intertidal zone, at 46%, while plastic blocks accounted for 40%. In the supratidal zone, plastic blocks had a higher proportion at 55%, while foam accounted for 38%. Microbeads and film-like microplastics had relatively low proportions in three sediment zones (Figure 3).

The size distribution proportions of microplastics in the three regions showed a similar trend, with the small microplastic particles (0.05–1 mm) comprising the majority of microplastics, and the proportions for bottom, intertidal, and supratidal zones were 85%, 91%, and 92%, respectively (Figure 4). White (including transparent and white) was the most common color of microplastics, accounting for 45%, 63%, and 59% of the total in the bottom, intertidal, and supratidal zones, respectively; the proportion of black was the second highest, while colored microplastics such as yellow, green, gray, and blue were relatively rare (Figure 5).

### 3.4. Composition of Microplastics in Beach Sediments on North Island, Qilianyu

A Fourier transform infrared spectroscopy analysis was conducted on 50 selected suspected microplastics from the aforementioned five categories. A total of 47 microplastic components were identified as plastics, as shown in the figure below. Among them, PE and PS accounted for the highest proportions, with 40% and 34%, respectively. The next highest proportions were observed for PP and PET, accounting for 21% and 4%, respectively (Figure 6 and Appendix A). The identification results indicated that the main components of plastic blocks were PE, PP, and PET. Both PET components in the plastic blocks were derived from the bottom zone, while no PET components were found in the plastic blocks from the intertidal zone. All foam microplastics were composed of PS, and the main components of fiber microplastics were PE and PP. PP was the main component of film microplastics. The proportions of the detected main components were similar to the findings of [29] in the Yongle Islands of the Xisha Islands region.

## 4. Discussion

Beaches are hotspots for the accumulation of microplastics and key areas of environmental pollution in the ocean [4,32]. Microplastic pollution is closely related to regional population activities and economic development [29]. In investigations and studies on microplastic pollution on beaches, a lack of uniformity in the surveyed particle size range is commonly observed, making it difficult to compare the abundance of microplastics between different regions. Therefore, this study only compared the abundance of microplastics on coastal beaches with similar surveyed particle size ranges to ours. The overall abundance of microplastics (size range: 0.05–5 mm) on North Island of Qilianyu was lower than in other places such as Guangdong and Hong Kong in China, possibly due to a smaller population and less urbanization, industrialization, and tourism-related activities in the area. However, since microplastics are stable and can persist in the environment for a long time, their abundance may increase over time. This characteristic requires our attention and measures should be taken to prevent an increase in microplastic pollution.

Most of the microplastics in Qilianyu Islands were white, followed by black. Similarly to the findings of [33], this may be the result of plastic weathering and fading in beaches or marine environments. The microplastics were mainly composed of small particles (0.05–1 mm), similarly to in previous research results [34]. However, the smaller the particle size of microplastics, the larger their specific surface area, which means they can absorb more pollutants, potentially causing greater harm to green turtle hatchlings [12]. Microplastics were widely present in the nesting grounds of Qilianyu Islands, even in close contact with turtle eggs [25]. The surface of microplastics can accumulate heavy metals and organic pollutants [16,17]. Ref. [35] showed the relationship between heavy metal contamination in the environment and that found in eggs. However, there is currently no research indicating heavy metals attached to the surface of microplastics can move into turtle eggs or whether microplastic particles can penetrate the eggshell membrane into the embryos. Therefore, further research and monitoring are needed in the future, such as the impact of microplastic accumulation on turtle incubation temperature and the effects of pollutants attached to the surface of microplastics on turtle incubation.

The types of microplastics in bottom sediment are mainly plastic blocks and fibers. This is mainly due to the large surface area and light weight of foams, films, and microbeads, which have a greater buoyancy and are not easily deposited on the seabed. Similarly to in the research results of [29] on other islands in the region, the main types of microplastics in the intertidal zone and supratidal zone were plastic blocks and foams. This is due to plastic blocks and foams being easily broken down by weathering and degradation, especially under high temperatures [30,31]. As tropical islands, Qilianyu Islands receive strong direct solar radiation, accounting for 60–70% of the total solar radiation, with an average annual temperature of approximately 27.4 °C [36], which is very suitable for plastic decomposition. Moreover, the beach temperature of Qilianyu Islands has been increasing year by year. From 2018 to 2021, it rose by 1 to 2 °C [25]. The breaking up of items such as plastic bottle caps into small plastic particles on the beach was commonly observed during the field work (Appendix A), indicating that large plastic debris break down on the beach of North Island and form small plastics or microplastics.

Furthermore, [37] detected microplastics in the seawater of this area and found that the main types of microplastics in the seawater of this region were fibers and films, with PET (56%) and PP (20%) being the main components. However, in this study, the microplastics extracted from the intertidal zone and supratidal zone were mainly plastic blocks and foam, with PS and PE being the main components, and no PET being found. Therefore, microplastics are unlikely to be transported from the seawater to the beach by waves. At the same time, the composition of microplastic debris was summarized, which mainly included PP, PE, PS, PET, PU, PVC, and EVA. Microplastic components in the intertidal zone and supratidal zone coexist with microplastics. Therefore, we believe that microplastics in the intertidal zone and supratidal zone of North Island in Qilianyu are likely to be the products of fragmented plastic debris on the beach.

Therefore, based on the above analysis, we believe that the microplastics on the beach of North Island are mainly generated from the gradual fragmentation and decomposition of large plastic debris on the beach, rather than directly from seawater. Beach cleaning activities that remove a large amount of plastic debris may be highly effective in preventing the generation of microplastics on the beach [31]. However, the current garbage cleaning on Qilianyu Islands mainly focuses on large plastic debris (size > 10 cm). Through regular cleaning efforts, the total amount of large debris has been controlled to a low level. However, there is still a large amount of small debris (1–10 cm) remaining after cleaning, and the cleaning effectiveness for small debris needs to be improved [24].

## 5. Conclusions

This study describes the level of microplastic pollution in sediments of North Island, where the abundance was 1513 ± 170 pieces/m^2^, which is similar to the microplastic abundance on Quanfu Island and Jinqing Island of the Xisha Islands, but much lower than that of coastal beaches in Guangdong and Hong Kong. The average abundance of microplastics in the bottom sediment was 277 ± 102 pieces/m^2^. In the intertidal zone, the average abundance of microplastics was 563 ± 194 pieces/m^2^, while in the supratidal zone, it was 2117 ± 398 pieces/m^2^. The average particle size of microplastics was mainly concentrated in the range < 0.05–1 mm. The most common color was white, with white foam being the predominant type, followed by black, while colored microplastics were less common. The main types extracted from the intertidal and supratidal zones were plastic blocks and foam, with the main components being polystyrene (PS) and polyethylene (PE). These were different from the composition and types of microplastics in seawater but were consistent with the composition of plastic debris on the beach. During the sampling process, we also observed the fragmentation of plastic debris on the beach. Therefore, we speculate that the main source of microplastics on the North Island is the fragmentation of plastic debris on the beach. It is recommended to further strengthen the regular cleaning of plastic debris on the beach, to reduce the threat to marine organisms such as sea turtles from microplastic pollution generated by the fragmentation of plastic debris.

## Figures and Tables

**Figure 1 toxics-12-00109-f001:**
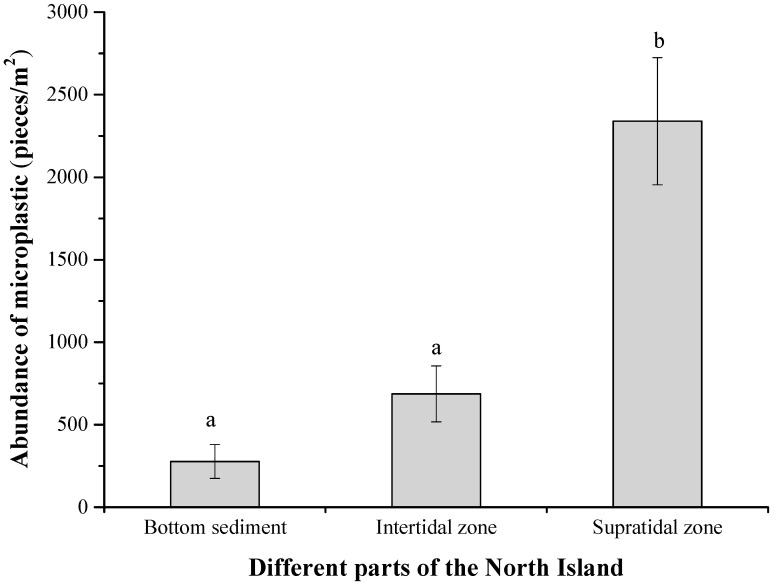
The abundance of microplastics in the bottom, intertidal, and supratidal zones of North Island, Qilianyu. Different lowercase letters in the figure indicate significant differences at the *p* < 0.05 level.

**Figure 2 toxics-12-00109-f002:**
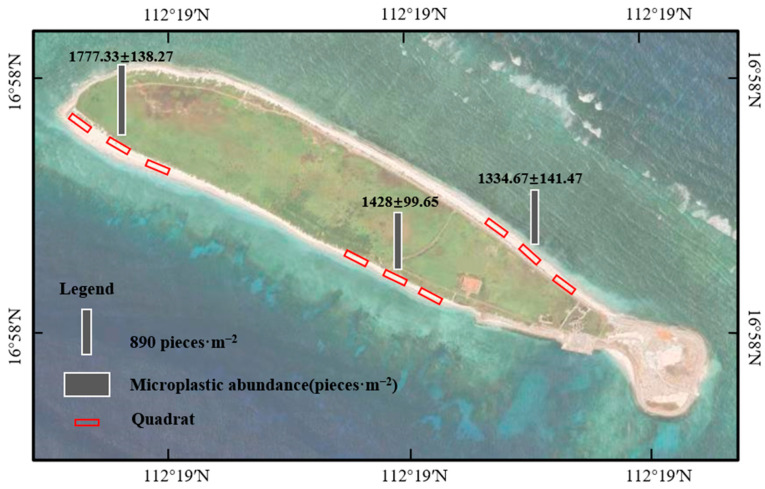
The abundance of microplastics in three transects on North Island, Qilianyu.

**Figure 3 toxics-12-00109-f003:**
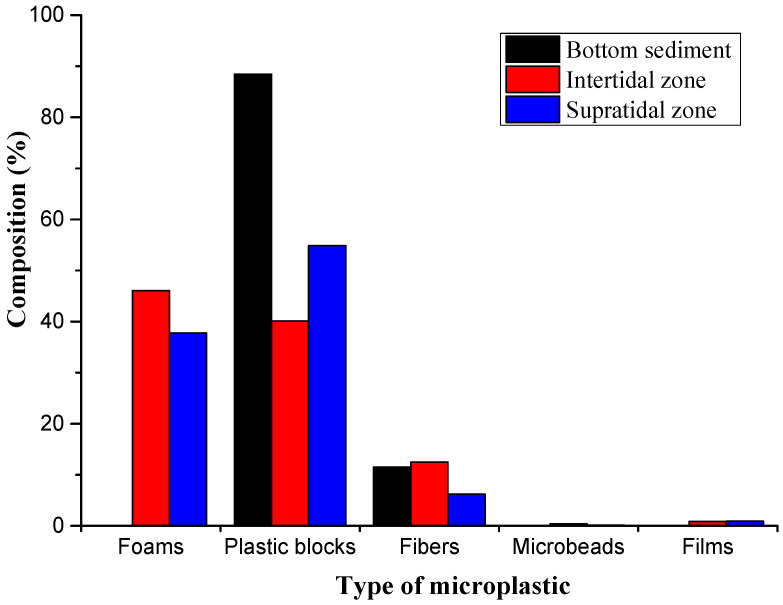
Types and proportions of microplastics in different areas.

**Figure 4 toxics-12-00109-f004:**
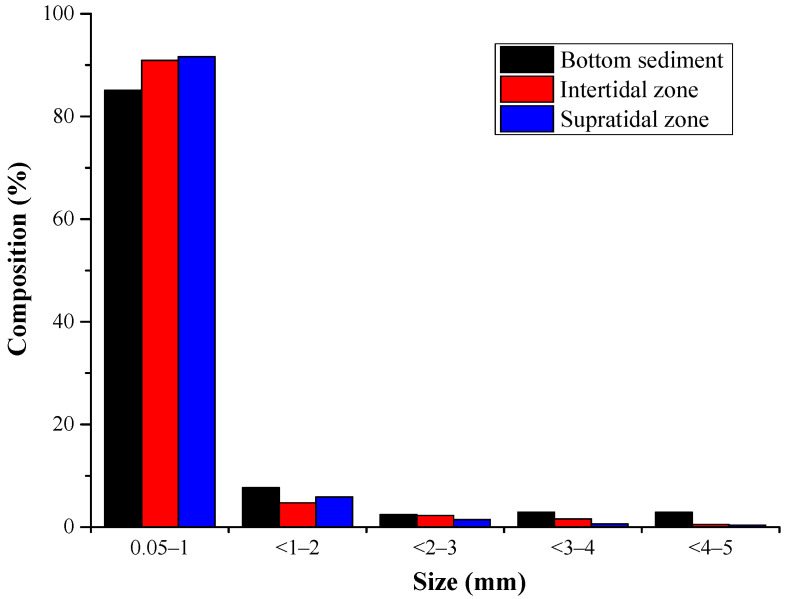
Size composition and proportion of microplastic particles in different regions.

**Figure 5 toxics-12-00109-f005:**
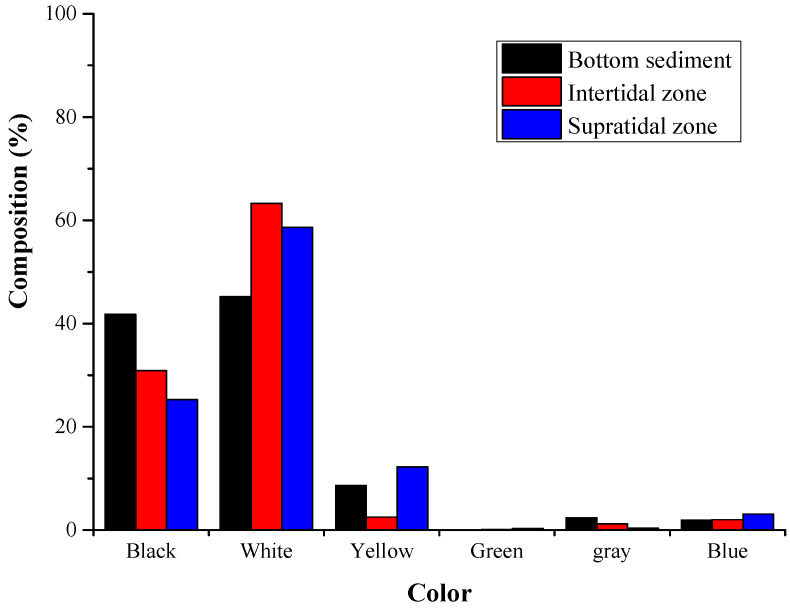
Color composition and proportion of microplastics in different regions.

**Figure 6 toxics-12-00109-f006:**
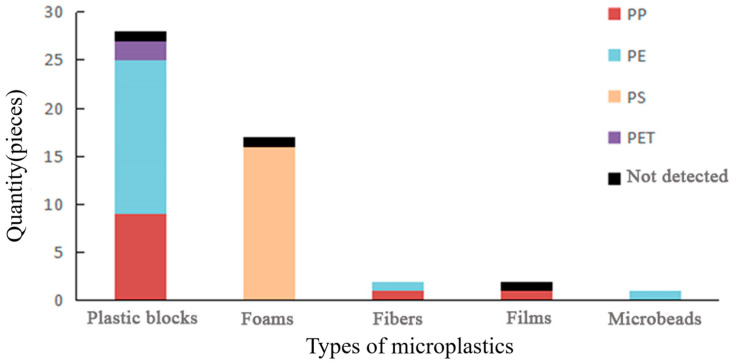
Main types and proportions of microplastics.

**Table 1 toxics-12-00109-t001:** Comparison of the abundance of microplastics in the intertidal and supratidal sediments of different sea areas.

Country	Study Area	Filter Membrane Aperture/μm	Abundance ± SD (Pieces·m^−2^)	References
China	North Island	45	1513 ± 170	In this study
China	Yongle Islands	20	1775	[29]
China	Hong Kong	315	5595	[30]
China	Guanggong Province	315	6675 ± 7021	[31]

## Data Availability

All data are available in the manuscript or in the respective Appendix A.

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
