# Peer review of "Microplastic Distribution Characteristics and Sources on Beaches That Serve as the Largest Nesting Ground for Green Turtles in China"

_toxics, 2024, doi:10.3390/toxics12020109_

Round 1

Reviewer 1 Report

Comments and Suggestions for Authors

It is an interesting manuscript shedding light into the levels and potential sources of microplastic pollution in the areas of the green turtle nesting sites. The topic is extremely relevant due to the harmful effects MP pollution poses to the hatchlings of the protected species, Chelonia mydas. I only have a few comments that the authors may wish to address prior to manuscript’s acceptance for publication.

Reviewer 2 Report

Comments and Suggestions for Authors

Manuscript: Toxics-2756909

Title: Microplastic distribution characteristics and source in beaches that served as the largest nesting grounds for green turtles in China

Authors: Ting Zhang, Deqin Li, Yunteng Liu, Yupei Li, Yangfei Yu, Xiaoyu An, Yongkang Jiang, Jichao Wang, Hai-Tao Shi and Liu Lin

Zhang et al. Have investigated the level of microplastic pollution in surface sediments from three different zones of the bottom sediment, intertidal zone, and supratidal zone on North Island, China.

The manuscript presents the microplastics number concentrations in an unstudied area, but otherwise the novelty of the manuscript Is not clearly defined. Based on but not limited to the comments below, the manuscript must go through the major revision.

General comments:

1.     The strength of the manuscript is that the particle sizes are clearly indicated unlike in many papers. On the other hand, the materials and methods should be described in details.

2.     The structure of manuscript is satisfying, but the careful finishing is lacking.

3.     The quality assurance and quality control are not described in section 2. Were the field and laboratory blanks used? Were the recovery of the analysis method determined? How was the sample contamination minimized? A separate section should be included.

4.     The separate section of Conclusions is encouraged to be included, which would highlight the findings of the present study.

Specific comments (Line numbers from the manuscript submitted):

1.     Line 14: Rephrase the sentence “Despite being far from the mainland, microplastic pollution cannot be ignored.” The message of the sentence is probably described in the lines 228 to 230.

2.     Line 56: The reference is published in 2011 and it is not based on the original data. Is there available a original paper that supports the statement?

3.     Line 58: “organic compounds” are very generally expressed, and they also include plastizers.

4.     Lines e.g. 81 and 89: It is a bit confusing since the same site is called as both southwest corner and southwest coast. Be precise with the terms.

5.     Lines 83 to 85: It is unclear which correlations are mentioned to be investigated. “Correlation between microplastics and microplastics on the beach” In addition, where the correlations are presented?

6.     Line 90: Respectively?

7.     Lines 93 to 94: “Each sampling point was repeated three times”?

8.     Line 104: Describe the container where the density separation was performed.

9.     Lines104 to 106: Could you explain why the density separation was firstly done in less dense solution of NaCl, which was followed the density separation of supernatant with more dense solution of NaI?

10.  Lines 107: What was the mixing ration of supernatant and hydrogen peroxide?

11.  Lines 115 to 122: Were collected microplastics really classified in total to 175 categories on a basis shape, colour and size? How about in the cases when the shape or color of microplastics was not possible to identify? In addition, how do the authors know which particles are plastic if the chemical analysis was not done before categorization?

12.  Lines 123 to 131: The description of the ATR-FTIR measurements is too vague. What are referred by “large particles”? Were the “small particle” (0.05-1 mm) not analysed at all? In addition to the model, give the manufacturer of the FTIR (Shimadzu?). What is the “ATR imaging attachment”? Were the ATR-FTIR measurements based on a single particle analysis? Were the particles really measured directly from the glass fibre filters? “Additionally, 50 random microplastic samples…” Addition of what?

13.  Line 133: MS EXCEL and version?

14.  Section 3: The measurement uncertainty does not support to present the results with even six significant numbers. Present the values with two significant numbers.

15.  Lines 147 to 152: These lines do not belong to Results. Move the lines to Discussion.

16.  Figure 1: Which column is for which site?

17.  Lines 173 to 175: Remove the lines since they do not belong to Results and are already described in section 2.

18.  Lines 178 to 179: It is unclear what are meant by “… but their proportions differ.”

19.  Lines 1181 to 183: These lines do not belong to Results. Move the lines to Discussion.

20.  Figure 3: Plastic blocks or fragments? Be precise with the terms.

21.  Lines 186 to 188: Check the grammar and revise.

22.  Figure 4: The unit of x axis is missing.

23.  Lines 194 to 202: The colors of microplastics have been presented in several papers but their information is usually minuscule. The results of colors are discussed with two lines (252 to 254). Remove the lines 194 to 202 and 252 to 254 or explain why this data is important and extend the discussion.

24.  Lines 204 to 206: 47 of 50 microplastics were identified as plastics, but what were the remaining three microplastics? Were they not plastics but still belonging to microplastics?

25.  Lines 211 to 212: Check and revise the term “the plastic fragment microplastics”.

26.  Figure 7: Remove or move to Supplementary Information.

27.  Line 224: “13 samples” and “7 samples”?

28.  Lines 252 to 254: See comment # 23.

29.  Line 260: Give the reference where the relationship between the MP abundance and beach temperature has been shown.

30.  Line 280: Remove “etc.”.

31.  Figures 8 and 9: Move to Supplementary Information.

Comments on the Quality of English Language

See the comments above.

Reviewer 3 Report

Comments and Suggestions for Authors

Rationale established for study (line 53). 

Line        Comment

45           More recent reference than Thompson et al. (2004)

46           define “native sources”

55           what’s the original ref for “microplastics on beaches can cause an overall increase in beach temperature?”  My gut feeling says the gender is more influenced by endocrine disrupting compounds adsorbed onto plastic

93           were these transects (e.g., Fig. 2) or point samples?  Were the locations chosen to be near nesting sites? 

153         I don’t believe the two decimals in Table 1, given the SD!!

160         Figure 1 is mislabeled?  Lines 158-160 appear to say that the abundances were from the difference intertidal elevations, but are all labeled bottom sediments.  And, post-hoc tests should follow ANOVA determination of significant difference among treatments.

171         why is abundance given units of (g m-2) month-1?

177        again, given the inherent variability, I’d round percent values to integers

182         “…which is similar to the research results of other islands in the same area investigated by Fang et al. (2020).”  This should be Discussion material

192         Dependent variable (composition) should have units of proportion, unless the numbers are converted to percent values.  I’d also suggest log-scaling the Y axis to better appreciate differences among sizes and locations.  Ditto for Figure 5.

218         Is Figure 7 necessary?

252         I’ve never been sure of the utility of color information

260         until I see results from a study that demonstrates the connection between microplastics and beach temperature, I remain very skeptical

269         maybe less the passage of microplastics into eggs than the movement of endocrine disrupting compounds.

284         Table 2 is superfluous (delete it); the information can be summarized in a sentence of text.

287         Figs. 8 and 9 belong in Materials and Methods

Round 2

Reviewer 2 Report

Comments and Suggestions for Authors

The responses of authors are very laconic "modified, thank you" but they have mostly made the satisfying revision of the manuscript. However, there are still comments that cannot be ignored:

"9) Could you explain why the density separation was firstly done in less dense solution of NaCl, which was followed the density separation of supernatant with more dense solution of NaI?

(Explain) The method of Thompson et al. (2004) is used for density separation, and sodium iodide with higher density is used to ensure the recovery of microplastics. We have added the reference at L111-112."

It remains unclear how the NaI increases the recovery of microplastics floating in NaCl. in addition, it is a bit misleading to cite to Thompson et al 2004, who only used a saturated NaCl solution in density separation, not a serial density separation as in the present study.

"23) Lines 194 to 202: The colors of microplastics have been presented in several papers but their information is usually minuscule. The results of colors are discussed with two lines (252 to 254). Remove the lines 194 to 202 and 252 to 254 or explain why this data is important and extend the discussion.

(Explain) Color is one of the important characteristics of microplastics, which can reflect the existence time and weathering degree of microplastics, especially white or transparent microplastics. We thought they should be kept, but reduced their information."

This is very questionable how the colour of microplastics can reflect the existance time and weathering degree of microplastics. The "aging" based on the colour has not been discussed in the paper. If the authors want insistedly save the colour data in the paper, they should discussed also "the important characteristics" or they should remove it from the manuscript. 

The authors added the following section on a basis of comment "2.4 Experiment quality control

All containers were rinsed at least three times with Milli-Q water and then dried before the start of the experiments. All plastic equipment was replaced with non-plastic if possible. If this was not possible, they were rinsed three times with Milli-Q water and then inspected to ensure that no plastic blocks were generated during sample processing. In addition, all containers were always covered with aluminum foil to avoid contamination. Nitrile gloves and cotton lab coats were worn throughout the experiment, with lab windows remaining closed (Zhang et al., 2022). Three procedural blanks were set to minimize contamination from the environment, and results showed that no microplastic particles were detected." It is susprising that no microplastics were found in three blank samples, even though the microplastics down to 50 micron was studied. How was the procedure blank prepared? Without sediment matrix but going through all the sample pretreatment steps?

The section of "data processing" should be titled as 2.5 (not 2.4). In addition, the data has beed processed with R, but the SPSS and Excel were mentioned in the original manuscript. Were the original information incorrect or is the data reprocessed with R?
